# OpenReview forum: "Beyond Induction Heads: In-Context Meta Learning Induces Multi-Phase Circuit Emergence"
_ICML.cc/2025/Conference — ICML 2025 poster_

### Official Review · Reviewer_CGRe · 2025-02-22

**Overall Recommendation:** 3

**Summary:**

This paper extends the toy model and task setup of [Reddy 2023] to the multi-task case, which requires models to infer tasks from in-context learning to label the last token.
Their main findings show that in this setup there are three phases in learning, where each phase is associated with the transition of an attention pattern/circuit in each of the two layers. This contrasts with the single phase for a single task as in [Reddy 2023].

They further investigate these findings and demonstrate:
- The three phases of learning persist across varying numbers of tasks, other than T=1 which results in only the first and last phase as in [Reddy 2023].
- The second phase circuit is unique to the T>1 IC-ML setup, and is associated with models’ improved accuracy by integrating label information to predictions. This circuit’s accuracy coincides with the accuracy of a model on tasks with random labels, suggesting that the reason model performance does not degrade fully on random labels is due to circuits such as this one.
- The phases are not pronounced with multi-head attention (as opposed to the single head setup), as the phased circuits form in parallel rather than in sequence.

## update after rebuttal

post rebuttal, raising my score to weak accept.

**Claims And Evidence:**

Yes they justify their experimental claims.

**Essential References Not Discussed:**

Not to my knowledge

**Experimental Designs Or Analyses:**

Yes, all checked. The numerical metrics used for the particular attention circuits are self explanatory and appropriate.

**Methods And Evaluation Criteria:**

If the focus is answering questions about their fixed setup, then I think the experiments/metric/evaluations make sense. However, the largest issue with the paper is that this toy task and model setup, which although reworked from prior art with [Reddy 2023], does not necessarily shed much light on in context meta learning in usual LLMs and transformers.
- The two layer, attention only models, and the last-token-only classification loss, require some leap of faith to infer meaning outside the toy setup to usual LLM’s trained with next token prediction. If the goal was to gain insight about LLMs outside this particular setup, there should have been other experiments carried out
    - Experiment training with data of varying context lengths and next token prediction, or at least label prediction at all $x$ tokens. Do the same circuits and phases form?
    - Experiments with more layers than two, and in a usual (single head architecture). Do the phases exist? What circuits arise?
    - Experiment with pre-trained foundation models performance on such a task, and perhaps ablate some of these circuits from all layers/attention heads.

**Other Comments Or Suggestions:**

post rebuttal, raising my score to weak accept.

**Other Strengths And Weaknesses:**

In addition to the weaknesses outlined in my answer to “Methods And Evaluation Criteria”, the paper extends the results of Reddy et al to the multi task setup, which detracts from the overall novelty and significance.

The strength of the paper is that the results they claim for their setup, are experimentally justified and well investigated.
Other than some under specified aspects of the setup, the clarity of the paper is good.

**Questions For Authors:**

- What are the 3 Tasks in T? I assumed it is a reassignment of labels and mean vectors for each class, but it could also be just a reassignment of labels, it is unclear.
- Appendix C does not explain the pruning/masking of circuits used.
- The setup is also not very clear for the “Deeper Look at the Semi-Context Circuit” section. Eg. “In Phase 1, since there are two tasks, the model has a 50% chance of predicting correctly by random guessing. In other words, the model’s prediction reduces to a binary choice for each input query $(x_q )$” Does this rely on the model having memorized the two possible labels that can apply to $x_q$ or does the input context contain all other classes so that possible 2 labels for $x_q$ can be decided by the two labels not used in the context? Can you clarify this section?

**Relation To Broader Scientific Literature:**

This would be most related to their referenced papers like [Reddy 2023], [Min et al 2022]

**Theoretical Claims:**

There was just one calculation, which determined the probability in the “Deeper Look at the Semi-Context Circuit” section. I checked it and do not find any issues.

---

> ### Author Rebuttal · Authors · 2025-04-01
>
> Thank you for your detailed review.
> We added experiments (notably Figures 21, 23, 24, 25, 26 and 27), in the following website: https://sites.google.com/view/in-context-meta-learning.
>
> **> W1**
> > Experiment training with data of varying context lengths and next token prediction
>
> We conducted additional experiments to address the concerns regarding generalizability.
>
> - Context Length:
> In **Figure 23**, we tested varied context lengths (N=2 to 64). While longer contexts accelerate convergence to the FCC, the phase transitions remain observable up to N=8. In **Figure 24**, we show the circuit learned for N=16, which structurally align with the FCC reported in the main text.
>
> - All-Label Prediction:
> In **Figure 25**, we experimented with predicting "all" labels in context. Phase transitions were still observed, and the final learned circuit again resembles the main FCC.
>
> These results demonstrate that the key phenomena—phase transitions and FCC formation—are robust to variations in context length and prediction objective.
>
> **> W2**
> > Experiments with more layers than two
>
> **Figure 26** shows experiments extending from 2-layer models to 3, 4, and 5-layer models. For the 2-layer and 3-layer models, we observe a phase transition in accuracy; however, when increasing to 4 or 5 layers, any phase transition in accuracy becomes less pronounced.
> Additionally, **Figure 27** visualizes the final circuits in the 3, 4, and 5-layer models. In all cases, the first two layers form a circuit resembling an FCC-like structure.
>
> Our results show that even with 3, 4, and 5 layers, the key circuit still forms, though phase transitions in accuracy become less distinct. This demonstrates that our findings extend beyond a mere two-layer “toy” setup, indicating a robust mechanism likely relevant to larger-scale models.
>
> **> W3**
> > Experiment with pre-trained foundation models
>
> Please see **W7** from Reviewer kgDv, where we show that a pretrained GPT2-XL exhibits chunk-example attention in its early layers and label-attention patterns in its middle-to-late layers on natural language data (see **Figure 21**). These findings align with our two-layer attention-only model and suggest that these circuits generalize to LLMs.
>
> **> W4**
> > the paper extends the results of Reddy et al to the multi-task setup, which detracts from the overall novelty and significance.
>
> We believe analyzing how Transformers operate in a more realistic few-shot in-context learning scenario is crucial, as few-shot performance is central to many practical LLM applications.
> While much of the existing circuit-focused research on in-context learning (e.g., induction heads) remains limited to copy tasks that are not seen in a practical LLM usage, our work explores circuit formation under a broader few-shot setup, offering a novel perspective.
> Furthermore, unlike the single-head approach in Reddy et al., we investigate multi-head architectures, shedding new light on how circuits differ in this setting.
> Finally, we also provide additional experiments with standard Transformers (see **W3** from Reviewer kgDv) and pretrained-LLMs (see **W3**), underscoring the broader relevance and novelty of our analysis.
>
> **> W5**
> > What are the 3 Tasks in T?
>
> As explained in Section 3.1, each task $\tau$ defines a unique assignment of labels $\ell$ to items $x$. While the underlying classes and mean vectors remain the same, the labels are randomly reassigned from one task to another. Thus, the 3 tasks in $T$ are simply 3 different ways of pairing each class (and its mean vector) with a label, giving rise to three distinct label assignments under the same class/feature space.
>
> **> W6**
> > Appendix C does not explain the pruning/masking of circuits used.
>
> We believe the pruning/masking procedure is described in Appendix C, where we state:
> ```all components except for the circuit corresponding to a specific phase were pruned at initialization, thereby isolating the contribution of each circuit.```
>
> In Figure 12, the legends Phase1 Circuit, Phase2 Circuit, and Phase3 Circuit each refer to an experiment where only the circuit from the respective phase was retained from the initial weights, ensuring that only that circuit contributed to the predictions. Due to the limited space in the response, we will provide additional details on the exact pruning process for each circuit in the revised paper.
>
> **> W7**
> > Does this rely on the model having memorized the two possible labels that can apply to xq
>
> It is clearly the case that the model memorizes the two possible labels corresponding to xq. As explained in Section 4.1:
> ```Phase 1 (Non-Context Circuit; NCC): Both layers use bigram attention, ignoring the context and relying solely on the model’s weights. ```
> During Phase 1, the model does not utilize any contextual information, indicating that the model’s weights have memorized the two possible labels for xq.

---

> > ### Comment · Reviewer_CGRe · 2025-04-02
> >
> > The additional experiments clear up most of my concerns; my only remaining lack of conviction comes from the utility of exclusively analyzing their generalized Reddy et. al. type setup. The added figure 21, analyzing the pretrained gpt-2xl is particularly interesting. Within that setup, their empirical results with the new additions are sufficiently convincing. I am changing my score to weak accept from weak reject.

---

### Official Review · Reviewer_bYju · 2025-03-12

**Overall Recommendation:** 3

**Summary:**

The paper proposes a problem setting named In-Context Meta-Learning (ICML) with multiple tasks.
For the same query, the answer would be different from task to task, so the model needs to infer the task to make a prediction.
Trained on this setting, the paper found that the model training has multiple phases: (i) predict simply based on query, (ii) predict based on labels of example and query, (iii) predict based on all.
The paper further shows multi-head attention may make this abrupt transition on loss not apparent, but such abrupt transition can still happen internally.

**Claims And Evidence:**

Claim: The model has multiple phase transitions: (i) predict simply based on query, (ii) predict based on labels of example and query, (iii) predict based on all.

Evidence: Almost all the experiments in the paper srtongly supports the claim.

Question: I believe Fig. 2 is used to support such claim. But the pattern is not obvious. The red block potentially make it even more hard to observe. I feel that label attention and bigram give the same visualizetion. (The case might be "this is just the truth". In that case, the author could consider draw the difference between the two visualizations of phase 1 and phase 2, or figure out another setting.)

**Essential References Not Discussed:**

Missing a very related work: Label Words are Anchors: An Information Flow Perspective for Understanding In-Context Learning. The paper empirically shows a very similar information flow: The label in the shadow layers collect information from the (x,y) pairs, and then deep layer processed those summarized information.

**Experimental Designs Or Analyses:**

Yes, I read the full paper so I checked the experimental design. I believe the paper should provide better writing to introduce the setting of the experiments. Based on my experience, I believe there is no issue in the experimental design and experimental results.

**Methods And Evaluation Criteria:**

Yes. The experiments make sense except for the Fig. 2, where the attention visualization is not well supporting the claim.

**Other Comments Or Suggestions:**

N/A

**Other Strengths And Weaknesses:**

Strength:

(1) The experiment of random-label robustness of SCC is a good connection to existing work, which is a plus to me. This is a strong evidence of the difference between phase 2 and 3.

(2) The experiment design is good to me. Starting from single attention and extend to multi-head attention

(3) Overall the experiments make the training dynamic and interpretation very clear.

Weakness:

(0) The author could show multiple run on experiments with multiple attention heads in Fig. 8. My training (multi-head) on ICL (different tasks) encounter diverse transition. The multiple plateaus may occur and may not. So I'm curious on whether the author constantly observe no transition happens on the accuracy curve.

(1) There are some minor issues in the flow of introduction:
(a) P1 col2 line051 "we train a simplified transformer", P2 col1 line101 "we also examine the case of multi-head model". I got confused here since the word "multi-head" suddenly jumped to me. I'll guess the conclusions before "multi-head" are based on a single-head model, then I'll guess whether the author means "single-head" as "simplified transformer";
(b) P2 col1 line101 "The existence .... nature of LLMs": The sentence is a statement but I think the statement may not be true since the paper found in the second phase the Transformer will predict based on examples' label (and query), while we do not know which phase real-world LLMs are in.
(c) P2 col2 line075 "phase transition can ... practical scenarios": this sentence says "bridging the gap". What is the gap between toy and practical settings?

(2) There is a redundancy in the related work:
(a) The first and second sentences aim to introduce the same thing “ICL”. I think the author forgets to delete one of them. (These duplicated sentences are potentially generated from LLMs, The author may need to do more proof reading.)

(3) There are some writing issue in the experimental setup:
(a) The notation $K$ is used in P3 col2 line 136 but not introduced before using. This issue introduces huge issue for me to understanding the paper. What is $K$? Why the author sets $L<K$? Is the $K=64$ the number of classes? Maybe not, since we only have $L=32$ labels? Such confusion from the issue costs lots of reading energy of me. The author should clarify in Sec. 3.1 rather than requiring the reader to remember the paper define $K$ in Fig. 1.
(b) $\mu$ is used in P3 col2 line130. It's used again in P3 col2 line149 for another meaning. The author could consider use \bm{\mu} for P3 col2 line130 since that's a vector.

(4) Experimental setting is not clear:
(a) Sec. 3.2 said there are $m$ heads, and more details in Appendix B. I went to appendix B but it does not tell me how many heads are used. I have this question because I want to know whether the results in Fig. 2 comes from 1 attention head or multiple attention heads. But I cannot find it. The question remains in my mind until I read the title of Sec. 5.
(b) P5 Col1 line253, "µ"-th layer.


I will rate 3.75 if purely based on experiments and ideas. The score goes to 3.25 because of the writing issues. (The qualities of idea and experiments are beyond acceptance line but the quality of writing is below acceptance line)

**Questions For Authors:**

N/A

**Relation To Broader Scientific Literature:**

The paper provides a deeper understanding of ICL beyond prior finding of induction head and a deeper understanding of the training dynamic of ICL beyond [I cannot provide more detail here since I'm not sure about what is the SOTA understanding of ICL's training dynamic].

**Theoretical Claims:**

There is no theory.

---

> ### Author Rebuttal · Authors · 2025-04-01
>
> Thank you for your detailed review.
> We revised the figures and added experiments (notably Figures 17 and 22) in the following website: https://sites.google.com/view/in-context-meta-learning.
>
> **> W1**
> > The red block potentially make it even more hard to observe
>
> We replaced the red squares with red arrows in **Figure 17** and highlighted the updated caption in purple. Strong attention stands for entries that exhibit relatively high intensity in the row-wise softmax-normalized attention map. Note that the last query token often shows strong attention because it’s crucial for prediction.
>
> Figure 2 is intended to illustrate the three distinct training phases (based on accuracy and loss) "qualitatively". This intuition from attention patterns motivates deeper analyses in the later sections. A more quantitative and rigorous analysis of attention maps appears in Figure 4 of the main paper.
>
> **> W2**
> >  K is not clear to me based on Sec. 3.1.
>
> We will include the following table, which outlines the key hyperparameters and their default values, as well as a brief description of each parameter. We believe this format clarifies how the data is generated and represented in our experiments.
>
> | **Parameter** | **Description**| **Default Value** |
> |---|---|---|
> | $T$           | Number of tasks (types of $\tau$).  | 3 |
> | $K$           | Number of classes (types of $k$).  | 64 |
> | $L$           | Number of labels (types of $\ell$).  | 32 |
> | $N$           | Number of $(x, \ell)$ context pairs.  | 4 |
> | $\epsilon$    | Noise magnitude controlling intra-class variation.  | 0.1 |
> | $p_B$         | Probability that the query item $x_q$ appears identically within the in-context examples. | 0 |
> | $\alpha$   | Exponent for the power-law (Zipf) rank-frequency distribution over classes.  | 0  |
> | $\beta$       | Exponent for the power-law (Zipf) rank-frequency distribution over tasks.  | 0    |
>
> **> W3**
> > Missing a very related work
>
> Thank you for noting [1]. Their results in pretrained models align with ours, especially regarding how label tokens gather context for deeper layers. Similar chunk-example attentions have also been noted in LLMs [2].
>
> Unlike [1] and [2], we run controlled experiments to observe circuit formation during training.
> We will clarify this connection and highlight how our dynamic analysis complements their findings.
>
> [1] Label Words are Anchors: An Information Flow Perspective for Understanding In-Context Learning.
> [2] Revisiting In-context Learning Inference Circuit in Large Language Models.
>
> **> W4**
> > the author constantly observe no transition happens on the accuracy curve.
>
> In **Figure 22**, we show four training runs (seeds 0, 1, 2, and 3) of a 2-head attention model. Consistently, across all seeds, the accuracy curve does not exhibit clear, discrete phase transitions, supporting our observation that multi-head configurations tend to smooth out in accuracy.
>
> **> W5**
> > I got confused here since the word "multi-head" suddenly jumped to me.
>
> > I went to appendix B but it does not tell me how many heads are used.
>
> We will clarify that our default setting uses a single-head attention model. We then introduce the multi-head extension later to examine how additional heads affect phase transitions and circuit formation.
>
> **> W6**
> > we do not know which phase real-world LLMs are in
>
> We agree that it is difficult to conclusively determine which “phase” might dominate within a real-world LLM. However, our point is that the label-attention circuit seen in Phase 2 of our toy model—one that yields higher accuracy on random labels—may also be present (at least in part) in large-scale models. We do not claim LLMs remain in phase2, only that phse2 circuit likely contributes to their observed behavior on random-label prompts.
>
> **> W7**
> > What is the gap between toy and practical settings?
>
> In our toy setup, we observe abrupt phase transitions (e.g., a sudden drop in loss), whereas large-scale LLM training typically does not exhibit such discrete transitions in practice. When we introduce multiple heads, these abrupt transitions become smoothed out, making our toy model’s behavior more akin to real-world LLMs—thus “bridging the gap” between the simplified setting and practical scenarios.
>
> **> W8**
> > Why the author sets L<K?
>
> Please see  **W2** for our detailed revisions to Section 3.1.
> We set $L<K$ following prior studies [1], which introduce a notion of synonymous classes: multiple tokens can represent the same label. For example, the words “Happy” and “Glad” might both map to the same sentiment label. This design reflects more realistic linguistic variability.
>
> [1] Data Distributional Properties Drive Emergent In-Context Learning in Transformers
>
> **> W9**
> > The author could consider use \bm{\mu} for P3 col2 line130 since that's a vector.
>
> We will modify the latter usage to indicate it is a vector, ensuring clarity in the notation.

---

> > ### Comment · Reviewer_bYju · 2025-04-01
> >
> > As in "Thank you for your rebuttal."

---

> > > ### Author Response · Authors · 2025-04-09
> > >
> > > Thank you for taking the time to review our paper and for acknowledging our rebuttal.
> > > If you have any further questions or concerns, we would be happy to address them.

---

### Official Review · Reviewer_d11A · 2025-03-13

**Overall Recommendation:** 4

**Summary:**

The main contributions of this paper are the following.

1. A novel synthetic in-context sequence modelling data set, based on identifying which of a number of classification rules are active and using it to predict the label of the query.
2. Demonstrating through analysis of the loss and attention mechanisms that
   when a small transformer is trained on this task, it progressively
   implements three different solutions of increasing accuracy over training.

Further experiments contribute additional findings, namely:

* One of the three solutions is a novel label-based prediction method, the
  paper studies this solution in more detail.
* The paper experiments with varying some data-distributional properties,
  cataloguing their effect on the sequence of solutions.
* The paper experiments with multi-head attention, showing that different heads
  specialise to different solutions at the same time in this case, and the loss
  trends are smoothed out.

**Summary of my review:** Overall, I think this is a strong paper that makes a
valuable contribution. However, I have recommended 'weak reject' based on
some missing discussion of important related work. If the authors can amend
this I would be pleased to change my recommendation to 'accept'.

**Update:** The rebuttal show adequate summary of the relationship to prior work. I am changing my recommendation to 'accept' in the understanding that these comparisons would be included in the final version. See my follow-up comment for some minor further discussion.

**Claims And Evidence:**

The empirical claims are supported by clear and convincing evidence for the most part.

I didn't quite follow figure 6's discussion of random label accuracy. In particular, RLA is as high in the SCC phase as in the FCC phase. In the latter, the transformer is meant to be using a circuit that depends on the pairs rather than just the labels. It seems to me that the RLA should be higher during SCC than during FCC.

**Essential References Not Discussed:**

**Synthetic in-context learning tasks that capture meta-learning.**

The paper acknowledges and discusses prior work on what they call 'copy tasks'
and the emergence of induction heads. They adequately compare their setting and
findings to this part of the literature on synthetic examples of the emergence
of in-context learning.

However, the 'copy task' setting is only a small part of a much broader
literature that has developed over the last couple of years, that is not
adequately discussed in the paper. There are by now dozens of different
synthetic in-context learning settings, many of which do not have the same
limitations as a simple 'copy task,' and are require more general requirement
of identifying a 'task vector' as discussed by the authors in the "implications
for LLMs" paragraph (starting line 424(right)). A non-exhaustive list of other
settings includes:

* In-context linear regression, which involves identifying a latent task
  vector, rather than just copying previous labels. The paper cites some work
  in this space but does not discuss it as an example of moving beyond the a
  'copy task.'
* In-context Markovian sequence modelling [1].
* In-context modular arithmetic [2].
* Many other similar settings.

In-context linear regression and Markovian sequence modelling involve
'implicitly' identifying a task vector. However within these settings there are
also variants more similar to the paper's setting where the transformer must
select from a finite set of tasks, namely [3, 4].

To be clear, the specific multi-task classification setting considered in this
paper is novel to my knowledge. However, since the paper frames this setting as
a core contribution and motivates it as an attempt to capture the challenge of
meta-learning, in my opinion related settings clearly represents related work
that should definitely be cited and ideally should be discussed in greater
detail.

**Multi-phase emergence.**

The other core empirical finding of the paper is that solutions arise in a
sequence of multiple phase transitions. While the analysis of this setting is
original (to my knowledge), a qualitatively similar finding has already been
published in the settings of Markovian sequence modelling [1], in-context
linear regression [5], language modelling [5], and plausibly other settings. I
believe this is highly-related work that should be cited and discussed in the
paper.

---

* [1] https://arxiv.org/abs/2402.11004
* [2] https://arxiv.org/abs/2406.02550
* [3] https://arxiv.org/abs/2306.15063
* [4] https://arxiv.org/abs/2412.01003
* [5] https://arxiv.org/abs/2402.02364

**Experimental Designs Or Analyses:**

N/A

**Methods And Evaluation Criteria:**

The synthetic data setting is well-designed and captures the intended features of a 'meta-learning' style task.

**Other Comments Or Suggestions:**

Small typos I noticed:

* Line 335: "nunber"
* Line 825: "Birstiness"

Other notes:

* The title of section 5, "multi-head enhances circuit discovery", was
  confusing while I read it. "Circuit discovery" to me suggests the mechanistic
  interpretability problem of finding circuits in a learned model. Importantly,
  it's mechanistic interpretability practitioners that are doing the
  discovering. However, this section appears to talk about the discovery of the
  circuits *by the model, during learning.* That is, multi-head attention makes
  it easier for different circuits to emerge. I invite the authors to consider
  a different title for this section to avoid possible confusion.

* On page 1, the paper uses a country-to-capital task to motivate moving past
  copy tasks and induction heads. While I generally agree with the framing that
  meta-learning is an important element of ICL, I didn't find this example
  compelling.

    Arguably, in a large-scale example, an induction head could be an important
  part of a mechanistic explanation of an LLM's ability to perform such a task.
  If by some intermediate layer the tokens `France`, `Spain`, and `Japan` have
  a representation involving that they are countries, and `Paris` and `Madrid`
  have a representation have a representation involving that they are capitals
  of the preceding country, then in this dimension the task becomes a copy
  task. The original induction heads paper already identifies and discusses
  similar 'abstract' induction heads in their small scale language modelling
  experiments (see the section 'Argument 4' in Olsson et al.).

**Other Strengths And Weaknesses:**

**Strengths.**

In my opinion, this is a strong paper overall. The results are very clear. The
presentation is very clear. I believe the paper makes a valuable contribution
to the literature, as discussed.

**Weaknesses.**

The main weakness is incomplete discussion of prior work, as outlined in the
previous section. To my knowledge, the setting proposed in the paper and the
specific results within this setting are novel, but as an alternative
demonstration of the same kinds of findings that I am already aware of based on
other work in different settings. I still think the paper is valuable and
well-done, but this prior work needs to be acknowledged and discussed.

**Questions For Authors:**

My overall assessment of 'weak reject' is based on the incomplete discussion of prior work. Do the authors agree that this work is related? If so, I invite the authors to clarify the distinction from this prior work in their revision, in which case I would be pleased to increase my recommendation to 'accept.'

**Relation To Broader Scientific Literature:**

Understanding the emergence of in-context learning abilities in large-scale
transformer models is an important objective of the science of deep learning.
Many recent works have studied simplified synthetic sequence modelling settings
where the emergence of in-context learning can be elicited and understood in
detail (in terms of both the emergent mechanistic structure and also the
training-dynamical mechanisms governing its emergence).

In this context, this paper's main contributions represent a well-designed and
novel synthetic in-context classification task that captures a meta-learning
challenge, exhibits a third intermediary 'semi-contextual' solution, leads to a
particularly crisp example of a progression of emergent learning mechanisms.
This setting would be a good basis for future work studying the emergence of
these transitions in detail.

**Theoretical Claims:**

N/A

---

> ### Author Rebuttal · Authors · 2025-04-01
>
> We appreciate the reviewers’ insights and address each point below.
>
> **> W1**
> >  It seems to me that the RLA should be higher during SCC than during FCC.
>
> Random Label Accuracy (RLA) can increase whenever label information is incorporated into the final token’s prediction. Even if the model uses a chunked example circuit (FCC), it can still leverage label information for that final prediction, which can maintain RLA. We do not claim that RLA must necessarily drop in the FCC phase or that RLA is definitively higher in SCC than in FCC.
>
> **> W2**
> > The main weakness is incomplete discussion of prior work
>
> We acknowledge that our study is related to multiple lines of research on in-context learning, multi-phase emergence. Below, we outline the connections and highlight where our approach differs.
>
> ---
> 1. ICL Literature Beyond Copy Tasks
> - **Linear Regression Approaches**
>   Prior studies on in-context learning often adopt linear regression [1,2], which provides a tractable theoretical framework. Although they set a common task across context examples, they typically rely on MSE loss—differing from real-world in-context use. Our setup more closely aligns with practical LLM applications.
>
> - **Markov Chain–Based Tasks**
>   Other works [3,4,5] focus on in-context learning with Markov chain tasks. While the meta-learning aspect is similar to our setting, these tasks differ from the few-shot pairs format that is standard in LLM applications.
>
> - **Modular Arithmetic**
> Research on in-context modular arithmetic [6] studies out-of-distribution behavior and final attention maps. Although it extends beyond simple copy tasks, it does not track the circuit acquisition dynamics during training or directly link them to phenomena like random-label accuracy or multi-head smoothing—key aspects of our work.
>
> ---
> 2. Multi-Phase Emergence Literature
> - **Phase Transitions**
>   Prior work [4] shows transformers acquiring functionalities in discrete “phase transitions,” which aligns with our observations. However, the task there (Markov chains) is different from our few-shot setup, which is more akin to real-world LLM usage. Also, while [4] studies a progression from `uniform prediction → unigram → bigram`, we analyze more complex `Bigram → Semi-Context → Full-Context` circuits.
>
> - **Developmental Interpretability**
>   Studies like [7,8] also note multi-phase transitions in in-context learning. However, they mainly focus on the loss landscape’s geometry (e.g., local learning coefficient, or LLC) rather than the mechanistic circuits we analyze. Bridging LLC-based approaches with an internal-circuit perspective could be an exciting future direction.
>
> We will include those discussions and citations in the revision.
>
> [1] Transformers Learn In-Context by Gradient Descent.
> [2] Pretraining task diversity and the emergence of non-Bayesian in-context learning for regression.
> [3] Selective induction Heads: How Transformers Select Causal Structures in Context.
> [4] The Evolution of Statistical Induction Heads: In-Context Learning Markov Chains.
> [5] Competition Dynamics Shape Algorithmic Phases of In-Context Learning.
> [6] Learning to grok: Emergence of in-context learning and skill composition in modular arithmetic tasks.
> [7] The Developmental Landscape of In-Context Learning.
> [8] Loss Landscape Degeneracy Drives Stagewise Development in Transformers.
>
> **> W3**
> > I invite the authors to consider a different title for this section to avoid possible confusion.
>
> We will rename that section to “Multi-Head Attention Facilitates the Emergence of Distinct Circuits” to avoid conflating our focus on circuit formation dynamics with mechanistic-interpretability efforts on circuit discovery.
>
> **> W4**
> > While I generally agree with the framing that meta-learning is an important element of ICL, I didn't find this example compelling.
>
> We agree that induction heads can perform pattern matching at a more abstract level, as described in the original induction heads paper (Argument 4 in Olsson et al.). However, consider a scenario where the token “Tokyo” does not appear in the context: if the model only performs a (possibly abstract) copy operation, it cannot generate “Tokyo” from the context itself. In other words, merely copying from the examples—whether literally or via a higher-level “abstract match”—is insufficient for a task like “Find the capital of Japan” unless “Tokyo” is explicitly present in the context.
>
> Thus, our central motivation remains unchanged: while induction heads can extract answers that are already present in the context, they do not fully explain the ability of LLMs to infer a shared task (like a country-to-capital mapping) and then apply it to a query whose answer is not directly provided in the context. Numerous studies (including research on task vectors) suggest that LLMs can learn and apply such tasks in-context beyond the scope of mere copy-based or induction-head operations.

---

> > ### Comment · Reviewer_d11A · 2025-04-02
> >
> > Thank you for your rebuttals. I changed my recommendation to 'accept' in the understanding that the proposed revisions (esp. comparison to related work) would be included in the final version.
> >
> > I briefly read the other reviews and rebuttals, and I am satisfied with most of the responses. I did not check the linked website with additional figures in any detail (I am not sure if sharing this extra information is allowed).
> >
> > **W4** One minor follow-up point regarding abstract induction circuits. Perhaps my example was unclear. I still believe an abstract induction circuit could implement this task even though "Tokyo" does not appear anywhere in the context. After an induction circuit in the intermediate layers of a language model literally copies the abstract continuation "capital of preceding country", subsequent layers could easily instantiate this as "Tokyo". I believe a similar circuit could explain many instances of meta-learning with no apparent copying. Anyway, I have no evidence for or against this proposal, and your claim in the introduction (that a *mere* induction circuit (without pre-/post-processing) cannot explain the ability) is technically true, so I am happy to leave it at that.

---

> > > ### Author Response · Authors · 2025-04-02
> > >
> > > Thank you for taking the time to review our paper, and we truly appreciate your updated score.
> > >
> > > Regarding the external link: as clarified in the [ICML 2025 Peer Review FAQ](https://icml.cc/Conferences/2025/PeerReviewFAQ),
> > >
> > > > While links are allowed, reviewers are not required to follow them, and links may only be used for figures (including tables), proofs, or code (no additional text). All links must be anonymous to preserve double-blind review, both in the URL and the destination.
> > >
> > > Our linked site contains only figures and follows the anonymity guidelines, so it complies with the ICML policy.
> > >
> > > **>  Re: W4**
> > >
> > > Thank you for the additional clarification.
> > > We now understand your point better—if the abstract pattern “capital of the preceding country” is represented in the intermediate layers and copied forward, it is plausible that an induction head could be part of such a mechanism. It’s a very interesting direction, and we appreciate you raising it.

---

### Official Review · Reviewer_kgDv · 2025-03-14

**Overall Recommendation:** 3

**Summary:**

The paper investigates how transformers acquire in-context learning (ICL) abilities by extending a simple copy task into an In-Context Meta Learning (ICML) setting that requires task inference rather than simply copying from the context. The authors train a two‐layer, attention-only transformer on a synthetic task where the model must infer the underlying task from (input, label) pairs and predict the answer for a query. A key contribution is the identification of three distinct learning phases—named the Non-Context Circuit (NCC), Semi-Context Circuit (SCC), and Full-Context Circuit (FCC)—each characterized by unique attention patterns (measured by metrics such as Bigram, Label Attention, and Chunk Example). The work further extends the analysis to multi-head settings and controlled pruning experiments, linking the observed circuit transitions to abrupt improvements in accuracy. Overall, the paper suggests that these emergent circuits underpin the meta-learning abilities observed in transformer-based language models.

**Claims And Evidence:**

**Claims**

- In-context meta-learning emerges via multi-phase transitions in the transformer’s internal circuitry.
- Each phase corresponds to a different circuit (NCC, SCC,FCC) which can be quantitatively tracked using specific attention-based metrics.
- These circuit transitions explain phenomena such as the model’s robustness to random label shuffling and the smoother improvement observed in multi-head settings.

**Evidence**

- Several experiments are performed on a simple transformer architecture which indeed exhibits the aforementioned transitions (fig 2, fig 4).
- The simple theoretical derivations (4.3), which yield predictions in line with empirical curves (fig. 5), support their interpretation of the circuits.


Some aspects (for example, the extension to practical large-scale LLMs) might benefit from further empirical validation beyond the synthetic setup as the considered architecture is a drastic simplification compared to modern LLMs.

**Essential References Not Discussed:**

From the best of the reviewer knowledge, the related literature is adequately covered.

**Experimental Designs Or Analyses:**

The experiments are extensive and aimed at showing that indeed the identified circuits emerge during training. I have a couple of questions:

1. Looking at the attention maps in Figure 2, it is not simple to identify the circuits. According to which criteria did the authors draw red squared in the attention maps? For example, it is not clear that in Phase 2, such squares actually highlight the entries of the matrix with high intensity. Also the figures are not very visible and in my opinion they should be made more visible and clear.
2. Is the accuracy calculated on test data?
3. The considered architecture is peculiar. Why did the authors choose to use 2 consecutive attention blocks followed by an MLP layer, rather than the classical transformer architecture interleaving attention and MLP blocks?
4. Figure 4. shows the evolution of the proposed metrics. These metrics follow to some extent the interpretation given by the authors. However, some metrics present some overlap which are not discussed in depth. For example, Chunk Example seems to be pretty high even in phases different from last one. Similarly, Bigram (layer 1) remains quite high in the phase. Do the authors have an explanation for that?
5. While the number of tasks is ablated in figure 7, it would be interesting to see what happens when T grows larger, a setting more in line with real world settings.

**Methods And Evaluation Criteria:**

The authors use mainly 4 metrics for their experiments: 1) accuracy and differential accuracy (as defined in 4.1), 2) bigram, 3) label attention, 4) Chunk example. The first is standard and its differential version makes sense to better illustrate  phase transitions in the dynamics, the others are specifically tailored to highlight the emergence of the identified circuits.

**Other Comments Or Suggestions:**

post rebuttal, raising my score to weak accept.

**Other Strengths And Weaknesses:**

**Strengths**

- Overall, the paper presents an interesting empirical analysis to investigate the phenomenon of in-context learning in transformer-based architectures.
- The learning dynamics revealed by the experiments is peculiar and novel from the best of reviewer knowledge.

**Weaknesses**

- Some experimental results should be more carefully discussed (see Experimental Designs Or Analyses).
- Some parts of the paper could be written more clearly, for example section 3.1 would benefit from a clearer explanation of the way the data is represented.
- It would be very interesting to see if the circuits found in the paper manifest themselves even in more standard architectures and in larger models.

**Questions For Authors:**

See Strengths And Weaknesses and Experimental Designs Or Analyses.

**Relation To Broader Scientific Literature:**

The paper builds upon prior work in in-context learning, notably studies on induction heads and mechanistic interpretability in transformers (e.g., Olsson et al., 2022; Reddy, 2023).

**Theoretical Claims:**

The paper includes a derivation of a theoretical accuracy formula for the SCC based on the probability that certain labels appear in the context (proof in Appendix D). Fig 5. empirically validates the theoretical result.

---

> ### Author Rebuttal · Authors · 2025-04-01
>
> Thank you for your detailed review.
> We revised the figures and added experiments (notably Figures 17, 18, 19, 20 and 21) in the following website: https://sites.google.com/view/in-context-meta-learning.
>
> **> W1**
> > attention maps in Figure 2, it is not simple to identify the circuits.
>
> Please see **Figure 17**, and refer to W1 from Reviewer bYju for more details.
>
> **> W2**
> > Is the accuracy calculated on test data?
>
> We focus on training dynamics and internal circuits in a practical task rather than OOD behavior as in [1]. Hence, we do not strictly separate train and test sets, although section 4.3 examines OOD behavior via random labels. For completeness, **Figure 18** presents test accuracy, confirming that phase transitions also manifest.
>
> [1] “The mechanistic basis of data dependence and abrupt learning in an in-context classification task”
>
> **> W3**
> > The considered architecture is peculiar.
>
> Using a two-layer standard Transformer, interleaving attention and MLP blocks, we still see phase transitions in accuracy and observe similar circuits (SCC,FCC) in **Figure 19**. However, the transitions are less distinct. We adopted a two-layer, attention-only Transformer to focus on self-attention, whose architecture is widely used in prior works analysing in-context learning [1,2,3]. The minimal models clearly simulates in-context behaviors such as induction heads [4], and can be useful for clearer phase-transition and circuit-level analysis.
>
> [2] “What needs to go right for an induction head? A mechanistic study of in-context learning circuits and their formation”
> [3] “Differential learning kinetics govern the transition from memorization to generalization during in-context learning”
> [4] “In-context Learning and Induction Heads”
>
> **> W4**
> > some metrics present some overlap which are not discussed in depth.
>
> We address why Chunk Example and Bigram remain high outside their respective phases:
>
> ---
> 1. Chunk Example
> Using the chunk-example metric (Table 2), the initial value is $\frac{1}{4}(\frac{1}{2}+\frac{1}{4}+\frac{1}{6}+\frac{1}{8})\approx 0.26,$which we treat as the baseline.
> - In **Phase 1**, both Layer 1 and 2 stay near this baseline.
> - In **Phase 2**, the metric increases in Layer 2 but does not affect the final output because the loss is only computed on the last token.
> - In **Phase 3**, the metric in Layer 1 exceeds the baseline and influences the final output, showing that the chunk-example circuit is forming and impacting the last token’s prediction.
>
> ---
> 2. Bigram
> - In **Phase 2**, due to row-wise softmax normalization, strong attention to label tokens in Layer 1 diminishes Bigram attention.
> - In **Phase 3**, in contrast, Layer 1’s chunk example pattern does not compete with Bigram, allowing the Bigram metric to recover relative to its Phase 2 level.
>
> We will add this explanation in the revised version.
>
> **> W5**
> > what happens when T grows larger
>
> We tested $T=3,6,9,12,15,18$ (see **Figure 20**) and found that phase transitions still occur for larger $T$. Increasing $T$ reduces Phase 1 accuracy (around $1/T$) and delays the transition to the next phase.
>
> **> W6**
> > section 3.1 would benefit from a clearer explanation
>
> Please see **W2** from Reviewer bYju.
>
> **> W7**
> > more standard architectures and in larger models.
>
> For standard Transformer results, please see **W3**.
> We tested whether our identified circuits appear in LLMs by evaluating a pretrained GPT2-XL (48 layers) on [SST2](https://huggingface.co/datasets/stanfordnlp/sst2) (872 samples). We used a 2-shot prompt: two labeled examples (`Review: {text}\nSentiment: {label}`) and a third, unlabeled query. Since the model is already trained, we observe each layer’s behavior on these prompts.
>
> We measure 3 metrics by averaging attention $p(i,j)$ across all heads in each layer, where $p(i,j)$ denotes the attention from the $j$-th token to the $i$-th token:
>
> 1. **Bigram**  $p(\text{query}, \text{query})$
>    Here, query refers to the final token, and this metric measures a query token’s attention to itself.
> 2. **Label Attention**  $\frac{1}{K}\sum_{k=1}^{K} p(\text{query}, \text{label}_k)$
>    Where $K$ is the number of shots, and $\text{label}_k$ is the label token in the $k$-th example. This captures how strongly the query token attends to label tokens.
> 3. **Chunk Example**  $\frac{1}{K}\sum_{k=1}^{K} \frac{\sum_i p(\mathrm{label}_k, \mathrm{text}_k^i)}{n_k}$
>    Here, $n_k$ is the number of text (non-label) tokens in the $k$-th example, and $\mathrm{text}_k^i$ represents the $i$-th such token. This metric reflects how strongly each label token attends to the text tokens in its corresponding example.
>
> **Figure 21** shows that the Chunk Example metric is higher in early layers, while Label Attention dominates later layers—mirroring our two-layer model’s progression from chunk example to label focus. This pattern aligns with our earlier findings in small Transformers, suggesting these circuits generalize to LLMs.

---

> > ### Comment · Reviewer_kgDv · 2025-04-09
> >
> > Thank you for your revisions. I am generally satisfied by the authors' rebuttal. I will raise my score to weak accept.

---

### Decision · Program_Chairs · 2025-05-01

**Decision:**

Accept (poster)

**Comment:**

This paper investigates how mechanistic circuits emerge during the training of transformer models on in-context meta-learning tasks. By analyzing a synthetic task, the authors identify multiple training phases, each characterized by the emergence of distinct attention-based circuits that support increasingly complex forms of generalization. The work is situated within the growing literature on mechanistic interpretability and in-context learning, and reviewers appreciated the empirical care and conceptual clarity of the circuit dissection—particularly the use of pruning and phase-specific interventions.

The main strength of the paper is its detailed, empirical account of circuit emergence in a controlled setting. While the analysis is task-specific by design, reviewers noted that it raises interesting questions about the dynamics of learning and representational shifts in broader settings. The paper does not attempt to establish generality, nor should it be expected to—but it would be helpful in the final version to explicitly discuss how future work might test the robustness of these findings across architectures or training setups.

One point of concern is the use of the term “phase transition.” While the training dynamics described are interesting and exhibit sharp shifts, referring to them as phase transitions is misleading. The observed changes occur in small models and finite settings, without the kind of asymptotic, large-scale behavior associated with non-analiticities of free energies that would justify such terminology. More neutral language such as training phase, or learning stage would more accurately reflect the empirical findings and avoid confusion. Clarifying this would strengthen the paper’s framing without diminishing its core contributions.

Overall, the paper offers a valuable empirical perspective on learning dynamics in in-context models. With clearer conceptual language, it makes a solid contribution to the interpretability literature and merits acceptance.